# Transition metal-doped Ni-rich layered cathode materials for durable Li-ion batteries

H. Hohyun Sun[1,6], Un-Hyuck Kim [2,6], Jeong-Hyeon Park[3], Sang-Wook Park[4], Dong-Hwa Seo [4], Adam Heller[1], C. Buddie Mullins [1,5✉], Chong S. Yoon [3] & Yang-Kook Sun [2✉]

Doping is a well-known strategy to enhance the electrochemical energy storage performance of layered cathode materials. Many studies on various dopants have been reported; however, a general relationship between the dopants and their effect on the stability of the positive electrode upon prolonged cell cycling has yet to be established. Here, we explore the impact of the oxidation states of various dopants (i.e., $Mg^{2+}$, $Al^{3+}$, $Ti^{4+}$, $Ta^{5+}$, and $Mo^{6+}$) on the electrochemical, morphological, and structural properties of a Ni-rich cathode material (i.e., $Li[Ni_{0.91}Co_{0.09}]O_2$). Galvanostatic cycling measurements in pouch-type Li-ion full cells show that cathodes featuring dopants with high oxidation states significantly outperform their undoped counterparts and the dopants with low oxidation states. In particular, Li-ion pouch cells with $Ta^{5+}$- and $Mo^{6+}$-doped $Li[Ni_{0.91}Co_{0.09}]O_2$ cathodes retain about 81.5% of their initial specific capacity after 3000 cycles at 200 mA g$^{-1}$. Furthermore, physicochemical measurements and analyses suggest substantial differences in the grain geometries and crystal lattice structures of the various cathode materials, which contribute to their widely different battery performances and correlate with the oxidation states of their dopants.

[1] McKetta Department of Chemical Engineering, The University of Texas at Austin, Austin, TX 78712-1589, USA. [2] Department of Energy Engineering, Hanyang University, Seoul 04763, Republic of Korea. [3] Department of Materials Science Engineering, Hanyang University, Seoul 04763, Republic of Korea. [4] School of Energy and Chemical Engineering, Ulsan National Institute of Science and Technology (UNIST), Ulsan 44919, Republic of Korea. [5] Department of Chemistry, The University of Texas at Austin, Austin, TX 78712-1589, USA. [6]These authors contributed equally: H. Hohyun Sun, Un-Hyuck Kim. ✉email: mullins@che.utexas.edu; yksun@hanyang.ac.kr

Lithium-ion batteries (LIBs) have attracted significant attention as power sources for contemporary electric vehicles (EVs). Among the various LIB components, the cathode is the most expensive and heaviest, and thus, it considerably influences the cost as well as the overall performance of a LIB; hence, the development of cathodes is critical to the success of LIBs.

The first transition metal (TM) oxide to be applied as a LIB cathode was $LiCoO_2$ (LCO)[1]. This material showed adequate electrochemical performance; however, due to its high cost, toxicity, and mediocre capacity, $LiNiO_2$ (LNO) was suggested as an alternative. Although LNO delivered much higher capacity at lower cost, it was unsuitable for commercial application owing to its inferior cycling and thermal stabilities, which were ascribed to the heightened surface chemical reactivity of $Ni^{3+/4+}$ and crystal structure destabilisation resulting from anisotropic internal strain caused by phase transitions in the deeply charged state[2–4]. Metallic dopants (such as Al, Ga, Mn, Mg, and Ti) were added to LNO for stability; however, they could not overcome the loss in capacity[5–9]. Therefore, mixtures of LCO and LNO, supplemented with a wide variety of tertiary elements ($Li[Ni_xCo_yM_z]O_2$ (M = doping element)), have been explored to realise cathodes with adequate capacity and cycling performance[10–12]. $Li[Ni_xCo_yAl_z]O_2$ (NCA) and $Li[Ni_xCo_yMn_z]O_2$ (NCM) layered cathode materials were obtained from these efforts, and they are currently the two most widely used layered cathode materials in commercial batteries[13].

Research on NCM and NCA layered cathodes has mainly focused on increasing the Ni content and, concomitantly, decreasing the Co content for the triple benefit of lower cost, higher specific capacity, and higher voltage, while maintaining cycling stability. Tremendous efforts have been devoted to this approach, and much progress has been made in successfully stabilising these cathodes, with Ni contents of up to 80%, for practical applications[14,15]. However, realising further improvements with this approach is challenging, as NCM and NCA cathodes with Ni contents exceeding 80% exhibit behaviours similar to those observed in LNO that undermine cycling and thermal stabilities. These instabilities are caused by the formation of intergranular microcracks, which serve as channels for deleterious electrolyte infiltration of the cathode particle interior and expedite the degradation of interior primary particles by reacting with the formed unstable $Ni^{4+}$ ions[16–18]. The microcracks originate from randomly oriented grains in cathode particles, which non-uniformly distribute the anisotropic internal strain caused by the abrupt collapse of the layered structure during phase transitions, particularly in highly delithiated states. Thus, engineering the geometry of cathode particle grains to dissipate strain build-up is vital to the cycling stability of Ni-rich layered cathodes.

In this regard, the conventional stabilisation strategy of doping the crystal structure fails to adequately address the problematic origins of the degradation mechanism as these dopants do not alter the random orientation of the equiaxed grains;[19–22] they merely delay the onset of decay. However, recent studies on B-, W-, Ta-, and Sb-doped cathodes have offered promising countermeasures against degradation[23–26]. By reshaping the primary particles of the cathode into radially aligned rod- or needle-like grains, the strain during cycling can be homogeneously distributed within the cathode particle to inhibit intergranular cracking and subsequent electrolyte penetration. Therefore, these Ni-rich cathodes demonstrate outstanding cycling and thermal stabilities, while retaining their characteristically high capacity and voltage. However, to date, the factors that differentiate the two classes of dopants have not been thoroughly investigated. In this study, we explore the distinguishing factors of dopants that optimize Ni-rich cathode materials through the introduction of Mg, Al, Ti, Ta, and Mo into $Li[Ni_{0.91}Co_{0.09}]O_2$ (NC90) cathodes and the investigation of their electrochemical, morphological, and structural properties. In particular, we investigate how the oxidation states of the dopants alter the geometry and crystal structure of the cathode grains to influence the cycling performance of LIBs, such that they can last the lifetime of an EV and subsequently be repurposed in energy storage systems (ESSs).

## Results

A model cathode delivers a high specific capacity and achieves an excellent cycle life, yet an intrinsic trade-off exists between these two features. As the graphite anode suffers no substantial capacity loss over thousands of cycles[16], the performance of a battery often depends on its cathode. To improve the performance of the NC90 cathode, we doped NC90 with 1 mol% of $Mg^{2+}$, $Al^{3+}$, $Ti^{4+}$, $Ta^{5+}$, and $Mo^{6+}$ to obtain cathode materials Mg-NC90, Al-NC90, Ti-NC90, Ta-NC90, and Mo-NC90, respectively. X-ray photoelectron spectroscopy (XPS) of the dopant parent materials, MgO, $Al_2O_3$, $TiO_2$, $Ta_2O_5$, and $MoO_3$, and corresponding doped cathode materials, Mg-NC90, Al-NC90, Ti-NC90, Ta-NC90, and Mo-NC90, confirm that the oxidation states of the dopant metals are +2, +3, +4, +5, and +6, respectively (Supplementary Fig. 1). Inductively coupled plasma optical emission spectroscopy (ICP-OES) of the synthesised cathode materials confirm their successful synthesis (Supplementary Table 1).

**Fundamental electrochemical performances**. The fundamental electrochemical performances of the NC90, Mg-NC90, Al-NC90, Ti-NC90, Ta-NC90, and Mo-NC90 cathodes, cycled in half cells in the voltage range of 2.7–4.3 V, are shown in Fig. 1a–c. The cycling Coulombic efficiencies are shown in Supplementary Fig. 2. The initial charge–discharge curves in Fig. 1a show that the cathodes deliver initial capacities of 227–230 mAh $g^{-1}$ (at 0.1 C and 30 °C, 0.1 C = 18 mA $g^{-1}$), which are typical of Ni-rich cathodes. Figure 1b shows that the cycling performances (at 0.5 C and 30 °C, 0.5 C = 90 mA $g^{-1}$) of the cathodes differ: the cathodes that are undoped and feature dopants with low oxidation states, $Mg^{2+}$ and $Al^{3+}$, retain 78.8, 82.5, and 83.7%, respectively, of their initial capacities after 100 cycles, whereas those featuring dopants with high oxidation states, $Ti^{4+}$, $Ta^{5+}$, and $Mo^{6+}$, retain 94.0, 97.0, and 94.9%, respectively, of their initial capacities. Note that the half-cell cycling performances of differing dopant concentrations (0.5, 1, and 2 mol %) of $Mg^{2+}$, $Al^{3+}$, $Ti^{4+}$, $Ta^{5+}$, and $Mo^{6+}$ in Supplementary Fig. 3 show that optimal cycling performances are obtained with a 1 mol % dopant concentration. Smooth and reversible phase transition peaks, especially in the case of the H2–H3 phase transition, are observed in the corresponding $dQ \ dV^{-1}$ profiles of these cathodes (Supplementary Fig. 4). Similarly, at a temperature of 60 °C, the cathodes that feature dopants with high oxidation states, $Ti^{4+}$, $Ta^{5+}$, and $Mo^{6+}$, retain 91.2, 93.5, and 93.1%, respectively, of their initial capacities, whereas the cathodes that are undoped and feature dopants with low oxidation states, $Mg^{2+}$ and $Al^{3+}$, retain 66.1, 80.1, and 80.2%, respectively, of their initial capacities (Fig. 1c). This difference in the cycling performances of the cathodes is amplified by long-term cycling in pouch-type full cells (with graphite anodes) in the voltage range of 3.0–4.2 V, at 1 C (200 mA $g^{-1}$), 100% depth-of-discharge (DOD), and 25 °C (Fig. 1d and Supplementary Fig. 5). The Mg-NC90 and Al-NC90 cathodes exhibit better cycling performances than the undoped NC90 cathode, yet cease to function effectively long before reaching 1000 cycles, achieving capacity retention values of 45.1 and 54.2%, respectively. To avoid rapid capacity loss, these cells should limit their DOD to 80%[27]. In contrast, the Ta-NC90 and

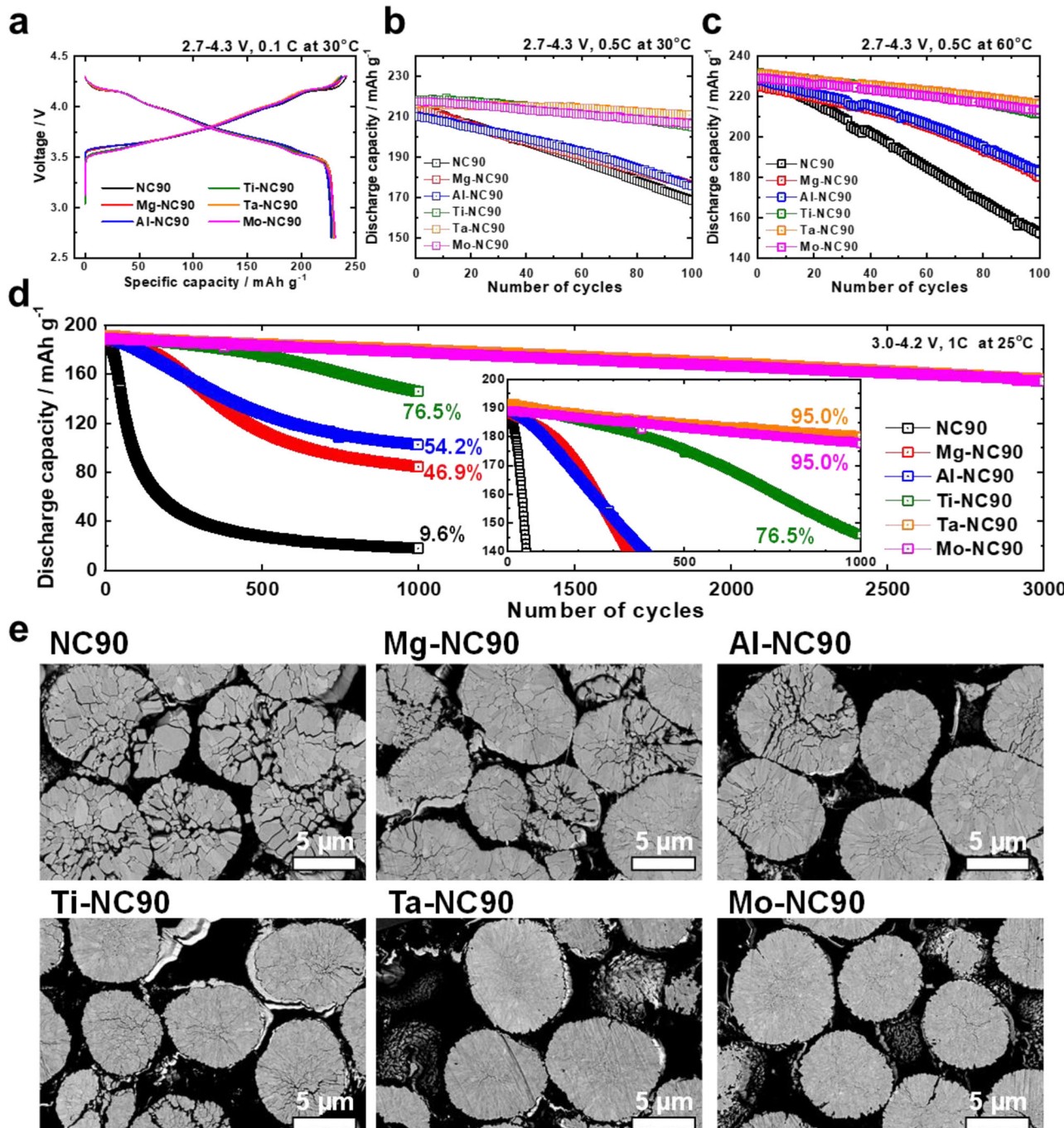

**Fig. 1 Cycling of NC90, Mg-NC90, Al-NC90, Ti-NC90, Ta-NC90, and Mo-NC90 cathodes in half cells within the voltage range of 2.7–4.3 V versus Li⁰/Li⁺ and full cells within the voltage range of 3.0 - 4.2 V versus graphite. a** The first charge–discharge cycle curves at 0.1 C (18 mA g⁻¹) and 30 °C. Cycling at 0.5 C over 100 cycles **b** at 30 °C, and **c** at 60 °C. **d** Cycling of NC90, Mg-NC90, Al-NC90, Ti-NC90, Ta-NC90, and Mo-NC90 cathodes in pouch-type full cells within the voltage range of 3.0–4.2 V versus graphite at 1 C (200 mA g⁻¹) and 25 °C. **e** Cross-sectional images of recovered NC90, Mg-NC90, Al-NC90, Ti-NC90, Ta-NC90, and Mo-NC90 cathodes in the discharged state of 2.7 V after 1000 cycles at 100% DOD.

Mo-NC90 cathodes demonstrate excellent cycling performances, retaining 95.0% of their initial capacities after 1000 cycles, and continue to deliver stable discharge capacities up to 3000 cycles. The final capacity retention value achieved by the Ta-NC90 and Mo-NC90 cathodes is 81.5%, verifying the stabilising efficacy of dopants with high oxidation states. Interestingly, the Ti-NC90 cathode, which performs similarly to the Ta-NC90 and Mo-NC90 cathodes in half cells, experiences accelerated capacity loss after 500 cycles in a full cell to retain 76.5% of its initial capacity after 1000 cycles.

The structural deterioration of the cathode particles correlates with their cycling performance, as revealed by the cross-sectional scanning electron microscopy (SEM) images of the cathodes recovered after 1000 cycles in full cells (Fig. 1e). The undoped NC90 particles are disintegrated into individual grains, and the Mg-NC90 and Al-NC90 particles display severely compromised particle integrity, with widespread microcrack networks extending along the grain boundaries to the particle surface. The Ti-NC90 cathode particles display less damage than the cathode particles featuring dopants with low oxidation states, but still

suffer from intergranular, hairline crack nucleation and propagation after extended cycling. In comparison, the Ta-NC90 and Mo-NC90 particles retain their particle coherency and present little to no signs of microcrack nucleation and propagation, which are consistent with their excellent cycling stabilities.

**Relationship between microstructure and electrochemical performance**. To investigate the influence of the oxidation states of dopants on the cycling and structural stabilities of cathode materials, the grain morphologies of the as-synthesised cathode materials were studied extensively. The SEM images shown in Supplementary Fig. 6 suggest that the precursor particles, $[Ni_{0.91}Co_{0.09}](OH)_2$, consist of thin, needle-like grains, which become thicker and denser when lithiated. At low magnification, the secondary-particle morphologies and sizes of the six cathode materials appear similar (Supplementary Fig. 7). However, SEM images of their cross sections (Fig. 2a and Supplementary Fig. 8) reveal noticeable disparities between the primary particles of the cathode materials: NC-90, Mg-NC90, and Al-NC90 grains are bulky, equiaxed, and vary in size, whereas Ti-NC90, Ta-NC90, and Mo-NC90 grains are thin, elongated, and align radially from the particle centre to its surface. During the 730 °C lithiation process, the needle-like primary particles of the precursors sinter into bulkier grains with the insertion of lithium atoms. The presence of Ti, Ta, and Mo dopants appears to preserve the original morphology of the precursor primary particles by suppressing sintering of the primary particles at 730 °C. To quantify the grain properties, the aspect ratio (expressed as length/width) and the acute angle between the longitudinal axis (a-axis) of each grain and the secondary-particle radial line passing through the centre of the particle were measured. Transmission electron microscopy (TEM) images, shown in Supplementary Fig. 9, confirm that the longitudinal axes of the grains are parallel to their a-axes, and illustrate the method by which the angle between the a-axis of each grain and the corresponding secondary-particle radial line was measured.

The results of the quantitative analysis of the cathode grain properties (i.e., the angle between the primary-particle a-axis and the secondary-particle radial line, aspect ratio, and size) are shown in Fig. 2b–d. The distributions of crystallographic orientation of the primary particles in Fig. 2b and Supplementary Fig. 10 indicate that a relationship exists between the oxidation states of dopants and the orientation of grains relative to the corresponding secondary-particle radial lines. The angles between the primary-particle a-axes and the corresponding secondary-particle radial lines of the NC90, Mg-NC90, and Al-NC90 grains range from −90° to 90° due to the random orientation of the equiaxed grains. As the oxidation states of the dopants increase, the distribution of angles narrows, indicating that the crystallographic textures of the Ta-NC90 and Mo-NC90 secondary particles are well-arrayed in the radial direction. Furthermore, the aspect ratios of the cathode grains tend to increase with increasing oxidation state of the dopant (Fig. 2c and Supplementary Fig. 10). The aspect ratios of NC90, Mg-NC90, and Al-NC90 are between 1 and 3, those of Ti-NC90 are between 3 and 4, and those of Ta-NC90 and Mo-NC90 are between 6 and 8. The grain width distribution also narrows with increasing oxidation state of the dopant, and the widths of the Ta-NC90 and Mo-NC90 grains, ranging from 50 to 150 nm, are less than those of the NC90, Mg-NC90, and Al-NC90 grains, which range from 200 to 800 nm. In a similar manner, the sizes of the Ti-NC90, and Ta-NC90 and Mo-NC90 grains at ~410 and ~310 nm, respectively, are smaller than those of the NC90, Mg-NC90, and Al-NC90 grains at ~650 nm (Fig. 2d).

The capacity retention values of the cathodes (after 1000 cycles) as functions of the average relative primary-particle angle, primary-particle aspect ratio, and primary-particle size are shown in Fig. 2e–g. The graphs indicate strong linear correlations ($R^2$) between capacity retention and the morphological characteristics of the cathode grains; low relative primary-particle angles, high aspect ratios, and small primary-particle size are strongly related to high capacity retention values after 1000 cycles. The intercept, slope, and $R^2$ of the regression lines are listed in Supplementary Table 2. Further examination of the angles between two adjacent grains of as-prepared cathode materials and corresponding cathode materials after 1000 cycles reveals that the angles between the adjacent grains of Mg-NC90 and Al-NC90 widen from the original ~30 to ~40° during long-term cycling, suggesting microcrack-induced grain separation, as illustrated in Supplementary Fig. 11. In contrast, the angles between adjacent grains in Ti-NC90, Ta-NC90, and Mo-NC90 cathode particles widen by only 5.9, 3.5, and 2.9°, respectively, during long-term cycling, owing to the preservation of particle integrity. Moreover, the results of micro-compression tests of particles of the six cathode materials demonstrate that the nano-sized grains of the cathode materials featuring dopants with high oxidation states substantially increase the resistance of the particles to mechanical stress (Supplementary Fig. 12).

The hybrid pulse power characterisation (HPPC) tests of the cathode materials, performed using pouch-type full cells, reveal that their initial direct current internal resistances (DCIRs) differ from their DCIRs after 50 cycles (Fig. 3a). The cathode materials were tested between 0 and 100% state-of-charge (SOC) at 10% intervals using the current pulse protocol proposed by the U.S. Department of Energy (DOE)[28]. During the 1st cycle, the initial DCIRs of NC90, Mg-NC90, and Al-NC90 cathodes are only slightly higher than those of Ti-NC90, Ta-NC90, and Mo-NC90 cathodes. After 50 cycles, the DCIRs of the cathodes that are undoped and feature dopants with low oxidation states are noticeably higher than those of their counterparts featuring dopants in high oxidation states across the entire SOC range. In particular, the increase in DCIRs of Ti-NC90, Ta-NC90, and Mo-NC90 is relatively low at 90% SOC (their ΔDCIRs are 43.6, 30.5, and 30.8%, respectively), which corresponds to the voltage range of the strenuous H2–H3 phase transition. In contrast, the DCIRs of NC90, Mg-NC90, and Al-NC90 are drastically higher at 90% SOC (their ΔDCIRs are 125.3, 110.4, and 105.1%, respectively). Correspondingly, the cross sections of NC90, Mg-NC90, and Al-NC90 cathodes recovered after 50 cycles and charged to 4.5 V show extensive microcrack networks, unlike the cross sections of the near-pristine Ti-NC90, Ta-NC90, and Mo-NC90 cathodes (Fig. 3b). This is significant because such microcrack networks undermine the electrical connectivity of individual grains and serve as channels for electrolyte penetration of the particle interior to trigger the formation of an insulating NiO-like rock-salt phase along the microcracks, further impairing the electrical conductivity of the cathode[29]. Similar trends are observed in the electrochemical impedance spectroscopy (EIS) measurements (Supplementary Fig. 13) using alternating current (AC). The charge-transfer resistance ($R_{ct}$) of cathodes featuring dopants with high oxidation states remain below 12 Ω over 100 cycles, but the $R_{ct}$ of NC90, Mg-NC90, and Al-NC90 continuously increase over the course of 100 cycles to reach 89.1, 45.3, and 63.4 Ω, respectively. The EIS curves of the 1st, 25th, 50th, 75th, and 100th cycle of the cathodes suggest that dopants with high oxidation states are more effective at enhancing the durability of layered Ni-rich cathode materials than dopants with low oxidation states.

**Relationship between atomic structure and electrochemical performance**. However, the grain morphologies alone do not sufficiently explain the disparities between the long-term cycling performances of Ti-NC90 and its counterparts featuring dopants

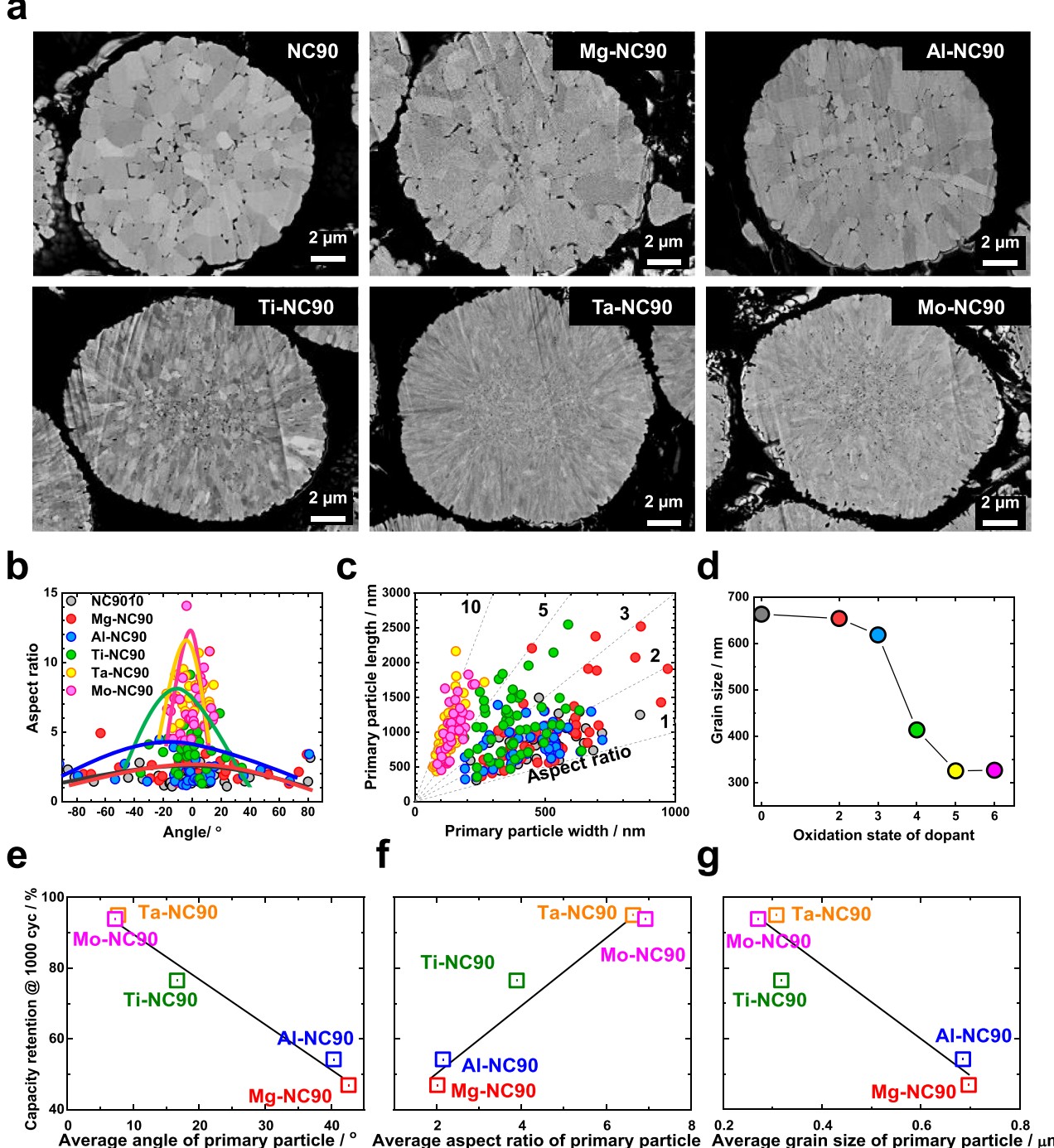

**Fig. 2 Correlation between primary particle morphology and full cell performance. a** Cross-sectional SEM images of as-synthesized cathode particles that show the morphology and orientation differences between the polygonal grains of NC90, Mg-NC90, and Al-NC90, and the elongated grains of Ti-NC90, Ta-NC90, and Mo-NC90. Quantitative analysis results of the cathode grains and their orientations. **b** Relationship between aspect ratio and relative grain orientation, **c** aspect ratio (length/width), and **d** grain size as a function of oxidation state. Summaries of the capacity retention values after 1000 cycles as functions of average **e** angle of primary particle, **f** aspect ratio, and **g** grain size.

with higher oxidation states. With this in mind, we investigated the crystal structures of the cathode materials; their Rietveld-refined, X-ray diffraction (XRD) patterns and determined structural characteristics are presented in Supplementary Fig. 14 and Supplementary Table 3, respectively. The calculated parameters of the cathodes, summarised in Fig. 4a, b and Supplementary Fig. 15, show that the $a$- and $c$-axis lattice parameters and unit cell volume increase monotonically with increasing oxidation state of

the dopant. Furthermore, during lithium extraction, the Ti-NC90, Ta-NC90, and Mo-NC90 cathodes experience lower volume changes than the NC90, Mg-NC90, and Al-NC90 cathodes, as illustrated by the results of in situ XRD measurements during initial charge (Supplementary Fig. 16). The Rietveld-refined XRD results also indicate that the degree of Li/Ni cation mixing in NC90, Mg-NC90, Al-NC90, and Ti-NC90 is relatively similar, but it increases considerably in Ta-NC90 and Mo-NC90, as

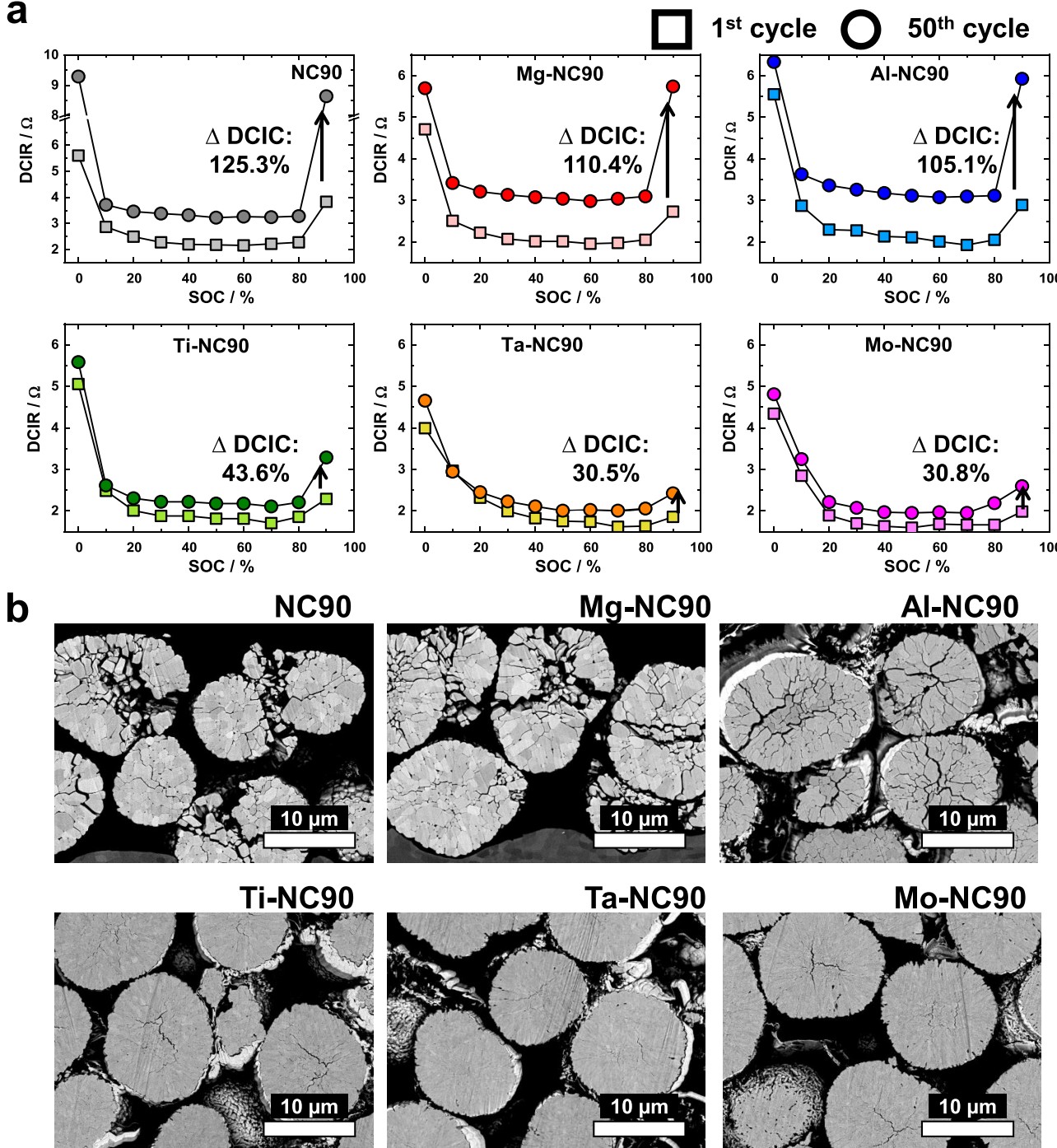

**Fig. 3 HPPC testing and mechanical stability. a** DCIR (direct current internal resistance) graphs of the six cathodes as a function of SOC at 1st and 50th cycle. **b** Cross-sectional SEM images of NC90, Mg-NC90, Al-NC90, Ti-NC90, Ta-NC90, and Mo-NC90, charged to 4.5 V after 50 cycles.

supported by the ratios of the intensities of their respective (003) and (104) Rietveld-refined XRD peaks (Fig. 4c and Supplementary Fig. 15). This is interesting because despite the general view that Li/Ni cation mixing is detrimental to cathode performance, the Ta-NC90 and Mo-NC90 cathodes exhibit appealing electrochemical performances. Likewise, the XPS analysis in Fig. 4d shows that significant proportions of Ni on the surfaces of the Ta-NC90 and Mo-NC90 cathodes have an oxidation state of +2, as indicated by the shift in their Ni $2p_{3/2}$ peaks toward lower binding energies (Supplementary Fig. 17 and Supplementary Table 4). We believe that the change in the oxidation state of proportions of Ni from +3 to +2 can be ascribed to charge neutrality

compensation, which is induced by the high oxidation states of $Ta^{5+}$ and $Mo^{6+}$ and is believed to cause the higher unit cell parameters and volumes of Ta-NC90 and Mo-NC90 ($r_{Ni}^{3+}$ = 0.56 Å and $r_{Ni}^{2+}$ = 0.69 Å)[30].

To determine the associated structural phase and its potential impact on the electrochemical performances of the cathodes, their crystal structures were examined by TEM. Figure 5a presents a typical high-angle annular dark-field (HAADF) image taken along the [100] zone axis of the Al-NC90 cathode material, which reproduces the atomic contrast expected from a perfectly layered lattice. The TM layer has a periodicity of 0.5 nm, whereas the Li layer is not visible owing to the low atomic scattering factor of Li.

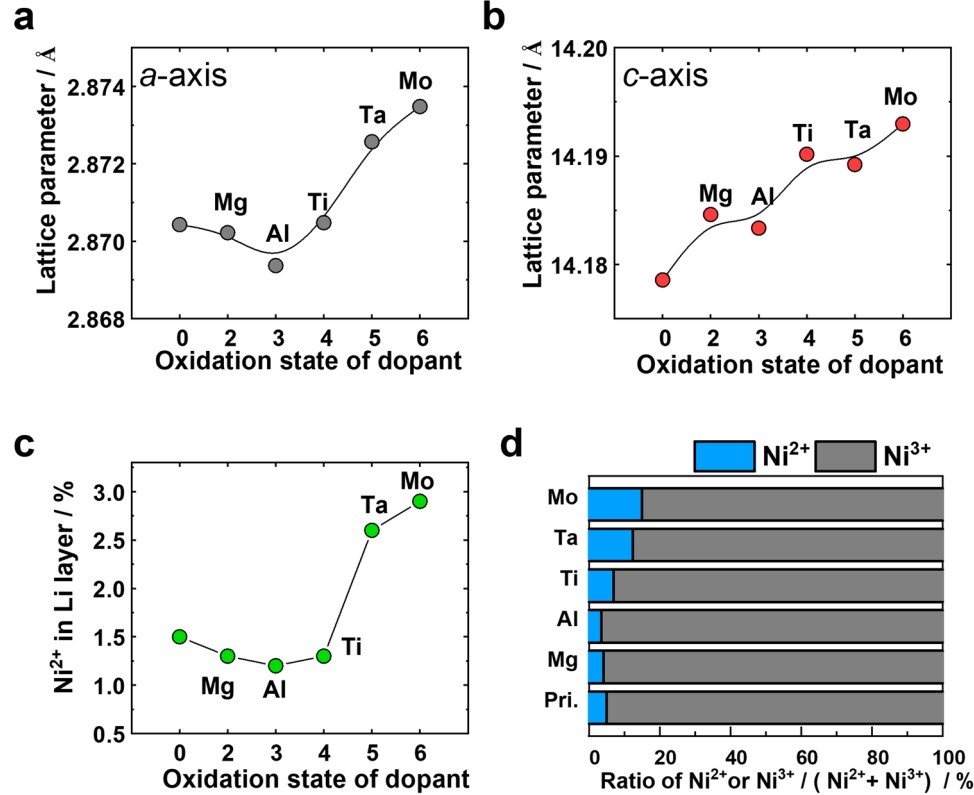

**Fig. 4 Summaries of crystal structures. a** $a$-axis lattice parameters, **b** $c$-axis lattice parameters, and **c** cation mixing through the interchange of $Li^+$ and $Ni^{2+}$ between layers. **d** The ratio of $Ni^{2+}/Ni^{3+}$ at the cathode surfaces as determined by XPS.

The HAADF images of the Mg-NC90 and Ti-NC90 cathode materials display similar patterns, suggesting that $Mg^{2+}$, $Al^{3+}$, and $Ti^{4+}$ negligibly affect the crystal structures of the cathode materials, which is consistent with previously reported results[31]. Conversely, the introduction of $Ta^{5+}$ and $Mo^{6+}$ into the NC90 layered lattice distorts it such that TM ions are detected in the Li layer in the HAADF images of the Ta-NC90 and Mo-NC90 cathode materials, and the TM layers exhibit periodic contrast (Fig. 5b). In previous studies[25,31], we demonstrated that their HAADF images display a superlattice structure formed by the ordering of anti-site defects where $Li^+$ ions alternately occupy the TM layer and vice versa. Figure 5c–g show a series of bright-field TEM images of primary particles for which the selected area electron diffraction (SAED) patterns were obtained along the [100] zone axis. The experimental SAED patterns of Mg-NC90, Al-NC90, and Ti-NC90 cathode materials match well with the SAED pattern of a typical layered structure, as shown in Fig. 5c, e, and g; however, the experimental SAED patterns of Ta-NC90 and Mo-NC90 cathode materials, shown in Fig. 5d and f, contain additional, regularly spaced spots (marked with yellow arrows). These additional spots are reflections created by the superlattice and match the simulated diffraction pattern in Fig. 5h, which was derived from the superlattice structural model. It should also be noted that the SAED aperture covers an area as large as 100 nm in diameter, indicating that Li/TM ordering is prevalent and occurs on a microscopic scale. In fact, nearly all the observed primary particles of the Ta-NC90 and Mo-NC90 cathode materials contain regions where ordered anti-site defects are observed. We believe that this superlattice structure stems from the high oxidation states of $Ta^{5+}$ and $Mo^{6+}$ that promote the partial reduction of $Ni^{3+}$ to $Ni^{2+}$ to maintain charge neutrality and lead to the migration of Ni into Li slabs with similar ionic radii. Unlike $Ta^{5+}$ and $Mo^{6+}$, it appears that $Ti^{4+}$ (at 1 mol%) is unable to

create observably large regions of Li/TM ordering. To verify the stability of the cation-ordered structures, the structural changes of cathode materials in highly delithiated states were directly observed by TEM. Even in the deeply charged state of 4.5 V, the ordered structure of the Ta-NC90 cathode is preserved, whereas the layered structure of the Al-NC90 cathode is severely damaged (Supplementary Fig. 18). Moreover, the cation-ordered structure of the Ta-NC90 cathode is still observed after 3000 cycles (Fig. 5i). This cation-ordering preserves the structural integrity of a cathode even when nearly all the Li ions are removed from the Li layer, thus enabling the long-term cycling of the cathode at 100% DOD. Without such a cation-ordered structure, the layered planes are more prone to collapse in highly delithiated states, thus creating irreversible structural damage (Fig. 5j).

**Li/Ni interlayer mixing energy calculation in crystal structure.** To confirm our hypothesis that the high oxidation states of dopants promote the transition from a layered to ordered structure, density functional theory (DFT) was employed to calculate the Li/Ni interlayer mixing energy ($E_{mixing}$)[32]. $E_{mixing}$ is defined as the energy difference between the perfect layered structure and the structure with one pair of Li/Ni interlayer mixing, where a low $E_{mixing}$ indicates preferred Li/Ni interlayer mixing. Within the monoclinic primitive supercell (space group: $P2_1/c$, $Li_{32}Ni_{31}MO_{64}$, M=$Ni^{3+}$, $Mg^{2+}$, $Al^{3+}$, $Ti^{4+}$, $Ta^{5+}$, and $Mo^{6+}$)[33], three neighbouring Li sites and all symmetrically distinct Ni sites were considered to be the most stable Li/Ni interlayer mixing configurations (Supplementary Fig. 19)[33]. The $E_{mixing}$ of the undoped, $Mg^{2+}$-, $Al^{3+}$-, $Ti^{4+}$-, $Ta^{5+}$-, and $Mo^{6+}$-doped structures are 0.896, 0.745, 0.778, 0.404, 0.144, and 0.075 eV, respectively (Supplementary Fig. 20), indicating that

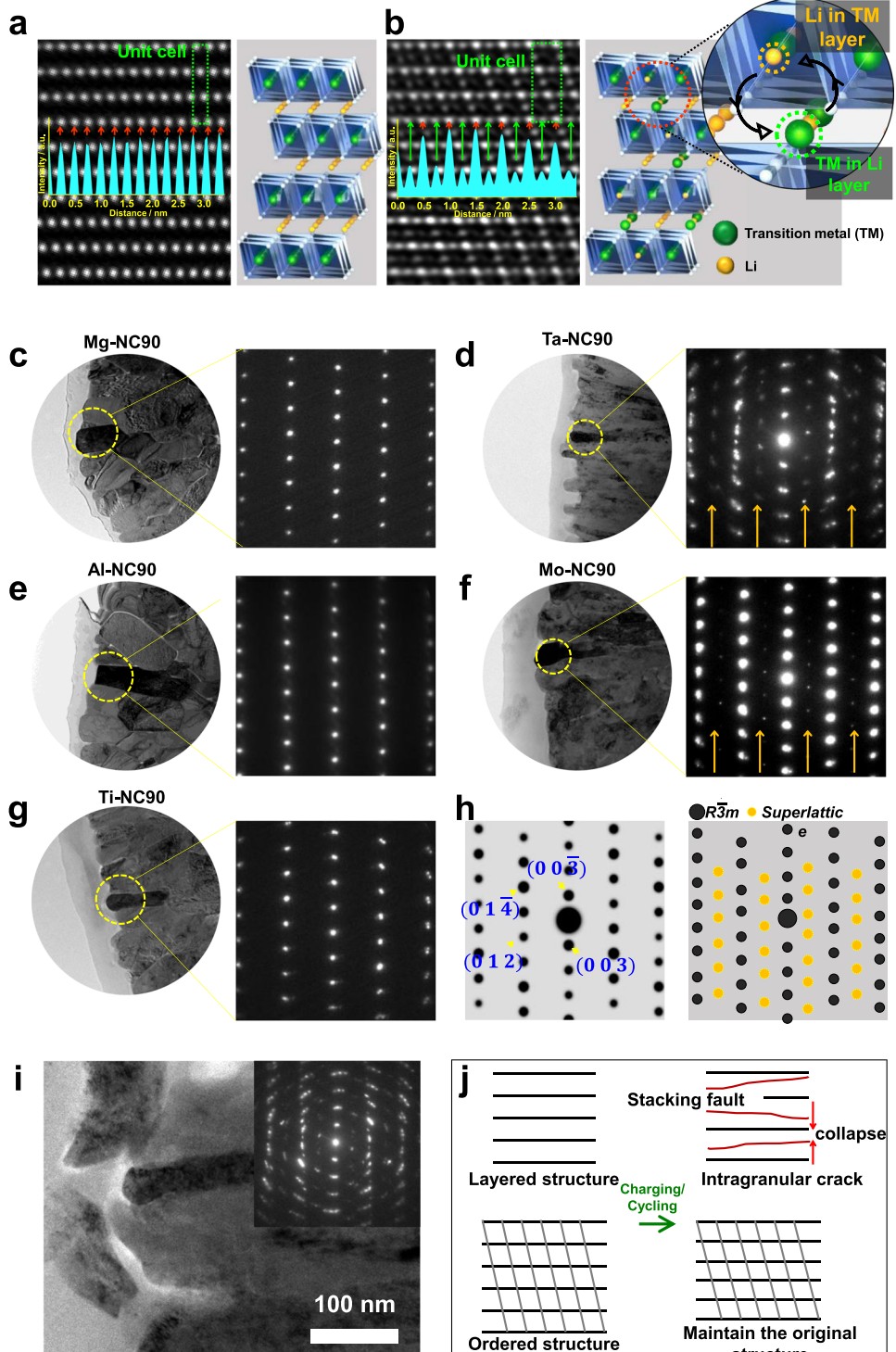

**Fig. 5 The high oxidation states of dopants promote the Li/TM ordered structure. a, b** HAADF TEM images and derived structural models of Al-NC90 and Ta-NC90, respectively. **c–g** Bright-field images and corresponding SAED patterns of single primary particles of Mg-NC90, Al-NC90, Ti-NC90, Ta-NC90, and Mo-NC90. **h** Calculated [100] zone axis diffraction patterns for the normal layered structure and ordered superlattice structure. **i** Post-mortem analysis of a discharged Ta-NC90 cathode after 3000 cycles using TEM. Low magnification TEM image of a cycled Ta-NC90 particle and corresponding electron diffraction pattern (inset). **j** Schematic comparing the relative structural stability of the normal layered and ordered structures in highly delithiated states and after long-term cycling.

$E_{mixing}$ decreases linearly with increasing oxidation state of the dopant. The low $E_{mixing}$ of $Ta^{5+}$- and $Mo^{6+}$-doped structures imply that interlayer mixing is favourable in such structures, which is consistent with our experimental observations in Fig. 4. The $E_{mixing}$ calculated using monoclinic $Li_{32}Ni_{31}MO_{64}$ (space

group: C2/m) with a different arrangement of elongated Ni–O bonds show a similar trend (Supplementary Fig. 20), indicating that Jahn–Teller Ni–O elongation does not affect $E_{mixing}$.

The magnetisation of Ni atoms and their local arrangement in $Li_{32}Ni_{31}TaO_{64}$ model structure with and without Li/Ni interlayer

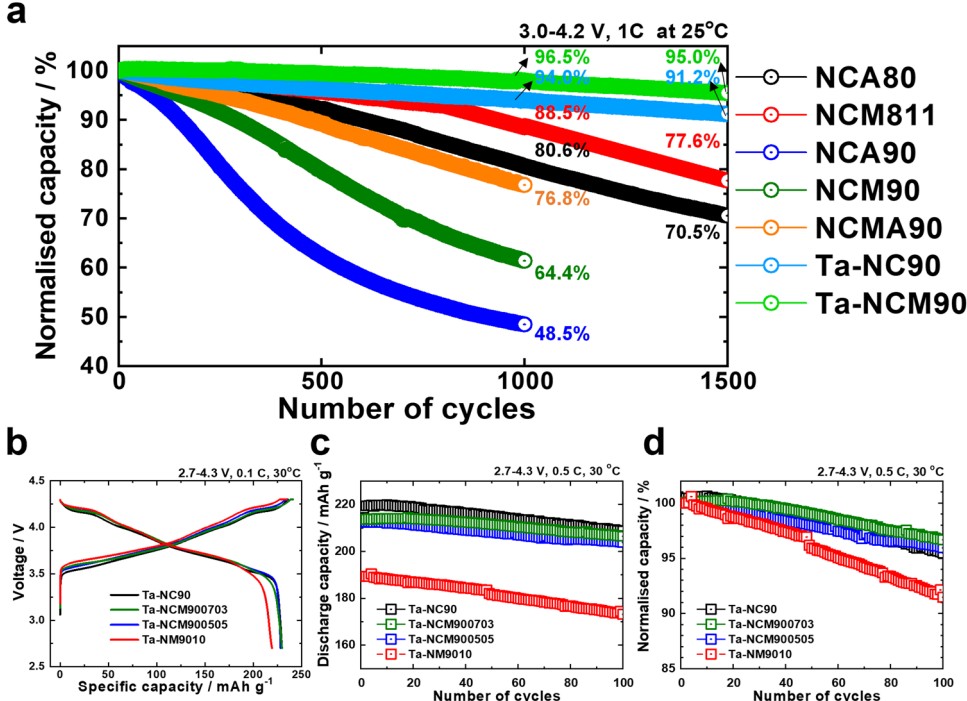

**Fig. 6 Optimized electrochemical performance via control of the cobalt to manganese ratio of the Ni-rich layered cathode. a** Comparison of the electrochemical performances of cathodes featuring randomly oriented primary particles and high Co content (NCA80, NCM811, NCA90, NCM90, and NCMA90); radially aligned primary particles, an ordered structure, and high Co content (Ta-NC90); and radially aligned primary particles, an ordered structure, and low Co content (4%) (Ta-NCM90); in pouch-type full cells. Cycling performances of Ta-NC90, Ta-NCM900703, Ta-NCM900505, and Ta-NM9010 cathodes in half cells within the voltage range of 2.7–4.3 V versus Li$^0$/Li$^+$: **b** first charge–discharge cycle curves at 0.1 C (18 mA g$^{-1}$) and 30 °C; **c** cycling over 100 cycles at 0.5 C (90 mA g$^{-1}$) and 30 °C and **d** normalised capacity.

mixing were also studied to elucidate the change in energetics arising from the reduction of Ni$^{3+}$ to Ni$^{2+}$. In the perfect layered structure of Li$_{32}$Ni$_{31}$TaO$_{64}$, Ta$^{5+}$ (0.0 μB) reduces the oxidation states of two neighbouring Ni ions from +3 (~1.1 μB) to +2 (1.72 μB), increasing the strain between Ni$^{3+}$ (0.56 Å) and Ni$^{2+}$ (0.69 Å) in the Ni layer and destabilising the Ta-doped perfect layered structure (Supplementary Table 5 and Supplementary Fig. 21). When the same structure is rearranged by interlayer mixing, two, strong linear Ni$^{2+}$–O$^{2-}$–Ni$^{2+}$ arrangements form between Ni$^{2+}$ (−1.71 μB) in the Li layer and Ni$^{2+}$ (~1.72 μB) in the TM layer to overcome the strain induced by the presence of Ta$^{5+}$ in the TM layer, resulting in the low $E_{mixing}$ of Li$_{32}$Ni$_{31}$TaO$_{64}$ (0.144 eV), as shown in Supplementary Fig. 21[34]. This linear Ni$^{2+}$–O$^{2-}$–Ni$^{2+}$ arrangement of Ni$^{2+}$ in the TM layer and Ni$^{2+}$ with opposite spin in the Li layer interconnected by O$^{2-}$ stabilises the structure through super-exchange interactions via σ-bonds formed between the $e_g$ orbitals of Ni$^{2+}$ in the Li layer and $p$ orbitals of O$^{2-}$ [34–36]. In the case of Li$_{32}$Ni$_{32}$O$_{64}$, the super-exchange interaction is not sufficient to compensate for the energy penalty associated with Li/Ni interlayer mixing, as suggested by the calculated $E_{mixing}$ of Li$_{32}$Ni$_{32}$O$_{64}$ (0.896 eV). Two linear Ni$^{2+}$–O$^{2-}$–Ni$^{2+}$ arrangements also form in Mo$^{6+}$-doped Li$_{32}$Ni$_{31}$MO$_{64}$ structure with Li/Ni interlayer mixing (Supplementary Fig. 22 and Supplementary Table 6), which explains its low $E_{mixing}$ (0.075 eV). Note that Ti$^{4+}$ (0.01 μB) does not reduce neighbouring Ni ions in Li$_{32}$Ni$_{31}$TiO$_{64}$ owing to charge delocalisation, making it difficult to form linear Ni$^{2+}$–O$^{2-}$–Ni$^{2+}$ arrangements. This is consistent with our experimental results which indicate that Ti$^{4+}$ does not induce the formation of an ordered superlattice structure.

Figure 6a and Supplementary Fig. 23 compare the normalised cycling performances of Ta-NC90 and its Co-poor Li[Ni$_{0.91}$Co$_{0.04}$Mn$_{0.05}$]O$_2$ counterpart (Ta-NCM90) to those of other layered cathodes (Li[Ni$_{0.80}$Co$_{0.16}$Al$_{0.04}$]O$_2$, Li[Ni$_{0.80}$Co$_{0.10}$Mn$_{0.10}$]O$_2$, Li[Ni$_{0.90}$Co$_{0.09}$Al$_{0.01}$]O$_2$, Li[Ni$_{0.90}$Co$_{0.05}$Mn$_{0.05}$]O$_2$, and Li[Ni$_{0.89}$Co$_{0.05}$Mn$_{0.05}$Al$_{0.01}$]O$_2$, denoted by NCA80, NCM811, NCA90, NCM90, and NCMA90, respectively). The cathodes were cycled in pouch-type full cells with graphite anodes, in the voltage range of 3.0–4.2 V, at 1 C, 100% DOD, and 25 °C. The capacities of both the NCA90 and NCM90 cathodes deteriorate rapidly and they fail to retain 70% of their initial capacities after 1000 cycles, demonstrating that simply increasing the Ni contents of NCA and NCM cathodes to 90% is not a viable approach to increasing their energy density. Although the cycling performance of the NCMA90 cathode is better than those of the NCA90 and NCM90 cathodes, its cycling stability is still inferior to those of the NCA80 and NCM811 cathodes. In contrast, the Ta-NC90 and Mo-NCM90 cathodes demonstrate appealing cycling performances, retaining 81.5% of their initial capacities after 3000 cycles. In addition to the cycling stability, the Ta-doped cathode is ideally located in the capacity retention (1000 cycles) versus specific capacity (1 C) plot in Supplementary Fig. 24. Other Ta-doped layered cathodes, which were intentionally designed with low Co contents, show similarly promising results (Fig. 6b–d and Supplementary Fig. 24). Evidently, the dopants with +5 and +6 oxidation states greatly stabilise the micro- and atomic structures of the cathode material to realise appealing electrochemical performances (under 100% DOD conditions) of the Ni-rich (>90%) layered cathodes in this study; these results may inform the commercialisation of Ni-rich (>90%) layered cathodes.

## Discussion

This work explores the impact of the oxidation states of dopants (Mg$^{2+}$, Al$^{3+}$, Ti$^{4+}$, Ta$^{5+}$, and Mo$^{6+}$) on the long-term cycling performance of layered Li[Ni$_{0.91}$Co$_{0.09}$]O$_2$ cathodes. The doped cathodes deliver different performances, where dopants with high oxidation states are more effective cathode stabilisers than those

with low oxidation states. Close examination of the primary-particle morphological characteristics and layered crystal lattices indicates that the stabilising effects of the dopants with high oxidation states manifest as highly oriented, elongated grain microstructures and Li/TM cation-ordered superlattice atomic arrangements. The former preserves the secondary-particle coherency to suppress increases in internal resistance and strongly correlates with capacity retention. The latter functions as atomic pillars to prevent the collapse of layered planes when deeply charged and is shown to be induced by +5 and +6 oxidation states that lower the Li/Ni interlayer mixing energy, as shown by DFT calculations. Hence, dopants with high oxidation states stabilise the layered cathodes, both at the micro and atomic levels, to last the lifetime of an EV. Future efforts should explore other dopants with high oxidation states that induce favourable features in cathode particle structures. Furthermore, efforts should focus on Ni-rich cathodes with Ni contents exceeding 90% to increase the energy density and decrease the Co content of LIBs.

## Methods

**Synthesis of spherical Ni-rich cathode materials**. This section describes the synthesis of spherical NC90, Mg-NC90, Al-NC90, Ti-NC90, Ta-NC90, and Mo-NC90 ($Li[Ni_{0.91}Co_{0.09}]O_2$, $Li[Ni_{0.90}Co_{0.09}Mg_{0.01}]O_2$, $Li[Ni_{0.90}Co_{0.09}Al_{0.01}]O_2$, $Li[Ni_{0.90}Co_{0.09}Ti_{0.01}]O_2$, $Li[Ni_{0.90}Co_{0.09}Ta_{0.01}]O_2$, and $Li[Ni_{0.90}Co_{0.09}Mo_{0.01}]O_2$, respectively). $[Ni_{0.91}Co_{0.09}](OH)_2$ precursors were synthesised according to the coprecipitation method using a solution of $2.0\,M\ NiSO_4\cdot6H_2O$ (aq) and $CoSO_4\cdot7H_2O$ (aq) (Ni:Co molar ratio = 91:9) as the starting reagents. A mixture of these reagents was fed into a batch reactor (17 L) filled with a solution of deionised water, $NH_4OH$ (aq), and NaOH (aq), under an inert gas atmosphere ($N_2$). Concurrently, 4 M NaOH (aq) (NaOH:TM molar ratio = 2:1) and $10.5\,M\ NH_4OH$ chelating agent (aq) ($NH_4OH$:TM molar ratio = 1.2:1) were separately pumped into the reactor. The final precursor powder was obtained by filtering and vacuum drying the product at 110 °C for 12 h. To obtain the various doped cathode materials, the precursor was mixed with $LiOH\cdot H_2O$ and the appropriate chemical compound (MgO, $Al_2O_3$, $TiO_2$, $Ta_2O_5$, and $MoO_3$) to achieve a Li:M:(Ni+Co) molar ratio of 1.01:x:1 − x, where x = 0 for NC90 and x = 0.01, for Mg-NC90, Al-NC90, Ti-NC90, Ta-NC90, and Mo-NC90. The powder mixtures were calcined at 730 °C for 10 h under an $O_2$ atmosphere.

**Physiochemical characterisations**. The chemical compositions of the prepared precursor and lithiated oxide powders were determined by ICP-OES (Optima 8300, PerkinElmer). A diffractometer (Empyrean, PANalytical) equipped with a Cu Kα radiation source was used to determine the crystal phases of the lithiated oxide powders. XRD data were obtained in the 2θ range of 10–110° with a 0.02° step size and analysed by applying the FullProf Rietveld refinement programme. In situ XRD experiments were conducted using pouch-type half cells with Li metal as the anode (Empyrean, PANalytical). The assembled pouch-type cells were charged to a cut-off voltage of 4.3 V at a constant current of 0.05 C ($9\,mA\,g^{-1}$). During the charging process, XRD patterns were recorded every 40 min using a detector (PIXcel 1D, PANalytical). The compression rupture strength of single particles was evaluated via micro-compression testing (MCT, MCT-W500, Shimadzu). The oxidation states of dopants in cathodes and relative $Ni^{2+}$ and $Ni^{3+}$ contents ($Ni^{2+}$ or $Ni^{3+}/(Ni^{2+}+Ni^{3+})$, expressed as a percentage) of the cathode surfaces were determined via XPS (Axis Supra, Kratos/PHI 5000, Versaprobe). Data was taken using a monochromatic Al-Kα X-ray source (15 kV, 300 W) with a spot size of $400\times400\,\mu m^2$. Dopant oxidation states in cathodes were measured with the electron flood gun on and the cathode surface was etched with a $Ar^+$ 500 eV ion etching gun prior to taking measurements. The C 1 s peak (284.8 eV) was used as a reference to calibrate the Ni 2p binding energy values. The particle morphologies and structures of the powders were observed by SEM (Verios G4 UC, FEI) and Cs-corrected TEM using a transmission electron microscope equipped with a cold field-emission gun (JEM ARM200F, JEOL). All TEM samples were prepared by using a focusing ion beam (FIB, NOVA 200/ FEI) with sample thicknesses being less than 100 nm for SAED work appropriateness. Cathode electrodes were cut using a cross-sectional polisher (IB-19520CCP, JEOL). The electrodes were harvested in Ar-filled glovebox and transferred into an Ar-filled atmosphere from cross-section polisher to SEM equipment using an Air-isolation system holder in order to minimize the air-exposure time.

**Electrochemical characterisations**. To fabricate the cathodes, the synthesised powders were mixed with a conducting agent, carbon black, and poly(vinylidene fluoride) (Solef 5130, Solvay) at a ratio of 90:5.5:4.5 (wt%) in N-methyl-2-pyrrolidone (NMP) using a mixer (ARE310, Thinky) at a high speed (2000 rpm) for 10 min. The slurries were coated on aluminium current collectors to achieve an active material

loading level of $4–5\,mg\,cm^{-2}$, vacuum dried at 110 °C for overnight, and roll-pressed to 3 kN. $1.2\,M\ LiPF_6$ in ethylene carbonate (EC)/ethyl methyl carbonate (EMC) (EC:EMC = 3:7 (vol%)) with 2 wt% vinylene carbonate (VC) was used as the electrolyte. The $H_2O$ content of the electrolytes was maintained at less than 20 ppm, as confirmed using a Karl Fischer titrator (C20, Mettler Toledo). Preliminary half-cell tests were performed using 2032 coin-type cells using Li metal (Honjo Metal Co., Ltd., Japan, Purity: >99.9%, thickness: 200 µm) as the anode and a polypropylene/polyethylene/polypropylene (PP/PE/PP, Celgard 2320, thickness: 20 µm) three-layer separator with 200 µL. The coin cells were charged and discharged at specific currents of 90 and $9\,mA\,g^{-1}$, respectively, at 0.5 C, 30 °C, and between 2.7 and 4.3 V (Toscat-3100, Toyo system). Long-term cycling tests were performed in laminated pouch-type full cells (55 mAh). For the fabrication of the full cell cathodes, the synthesised powders were mixed with carbon black and poly(vinylidene fluoride) at a ratio of 94:3:3 (wt%) in N-methyl-2-pyrrolidone. The obtained slurries were applied on carbon-coated aluminium foils to achieve an active material loading level of $9–10\,mg\,cm^{-2}$. Commercial artificial graphite (Posco Chem.) was mixed with a conducting agent, carbon black, and poly(vinylidene fluoride)(Soelf 5130, Solvay) at a ratio of 94:1:5 (wt%) in N-methyl-2-pyrrolidone (NMP) using a mixer (ARE310, Thinky) at a high speed of 1500 rpm for 10 min. The slurries were then coated on copper current collectors to achieve an active material loading level of $5–6\,mg\,cm^{-2}$, vacuum dried at 110 °C for overnight, and roll-pressed. The sizes of the positive and negative electrodes in the pouch-type cells were $3\times5\,cm^2$ and $3.1\times5.1\,cm^2$, respectively. The pouch cells were assembled with one double-side coated negative electrode and two single-side coated positive electrodes and contained the 0.8 mL electrolyte. Each layer between the positive and negative electrodes were wrapped with a separator (polypropylene/polyethylene/polypropylene (PP/PE/PP, Celgard 2320) three-layer separator), and a total of three layers were used. Based on areal capacity, the N/P ratio for the full cells was 1.15–1.20. Before cycling, the pouch cells underwent formation step for activation of cell and formation of stable solid electrolyte interphase (SEI) at 3.0–4.2 V, 0.1 C, and 25 °C and were subjected to degassing step. The full cells were charged and discharged between 3.0 and 4.2 V at a constant current of 1 C (55 mA corresponding to $200\,mA\,g^{-1}$) at 25 °C. All electrochemical tests were conducted under an environmental testing chamber to control the precise temperature (Temperature accuracy: ±0.2 °C). Electrochemical impedance spectroscopy measurements were performed using 2032 coin-type half-cells charged to 4.3 V at 30 °C by a multi-channel potentiostat (Bio-Logic, VMP3) over the frequency range from 1.0 mHz to 1.0 MHz and with a voltage amplitude of 10 mV. The number of data points from $f_i$ to $f_f$ is 151.

**Computational details**. First-principles calculations, based on DFT, were carried out using the Vienna Ab initio Simulation Package[37]. The Perdew–Burke–Ernzerhof functional[38] was adopted for the generalised gradient approximation (GGA) exchange-correlation. Hubbard U correction (GGA + U)[39] was introduced for Ni and Mo with U values of 6.2 and 4.38 eV, respectively[40]. A plane-wave cut-off energy of 520 eV and $6\times5\times6$ k-point meshes were used for the calculations. All the structures were fully relaxed until the forces on each atom were below 0.02 eV.

**Reporting summary**. Further information on research design is available in the Nature Research Reporting Summary linked to this article.

## Data availability

All the data generated or analysed during this study are included in this published article and its Supplementary Information. The data that support the graphs within this paper and other findings of this study are available from the corresponding author upon reasonable request.

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

## Acknowledgements
This work was supported by a National Research Foundation of Korea (NRF) grant funded by the Korean government Ministry of Education and Science Technology (MEST) (no. NRF-2018R1A2B3008794) (Y.K.S.). This work was also supported by the Human Resources Development programme (No. 20214000000320) of a Korea Institute of Energy Technology Evaluation and Planning (KETEP) grant, funded by the Ministry of Trade, Industry and Energy of the Korean government. (Y.K.S.). This work was also supported by the generous support of the Robert A. Welch Foundation via grant F-1436 (C.B.M.).

## Author contributions
H.H.S., U.H.K. and Y.K.S. conceived and designed the research. H.H.S., U.H.K. and J.H.P. performed the experiments and characterization of materials. H.H.S., U.H.K., C.S.Y. and Y.K.S. analysed the data. S.W.P. and D.H.S. conducted DFT calculations. H.H.S., U.H.K., S.W.P., D.H.S., A.H., C.B.M., C.S.Y. and Y.K.S. contributed to the discussion of the results. H.H.S., U.H.K. and Y.K.S. wrote the manuscript. All the authors commented and revised the manuscript.

## Competing interests
The authors declare no competing interests.
