## [Peer Review File · Nature Communications]

REVIEWER COMMENTS

Reviewer #1 (Remarks to the Author):

The paper submitted by Sun et al. demonstrated that the valence states of the dopants played a key role in enhancing the electrochemical properties of the NC91 materials. The dopants with diverse valence states can alter the morphological properties of the end particles in different levels. The dopants with a high valence state can strengthen the particles, thus leading to improved performance. It's clever that the authors calculated the orientation of grains in each doped particle based on the TEM technique to support their proposal. Besides the morphological viewpoint, a structural view is also claimed. The high valence state Ta⁵⁺ and Mo⁶⁺ can active the formation of superlattice structure, while the dopants with low valence state cannot. This research is complete and I believe it can provide helpful information in guiding the design in the Ni-rich cathode field. However, some points are still confusing.

Q1. I was a little confused that why the authors tested the valence states of the elements in dopant parent materials, rather than the final materials. There might be some differences between the valence states in their parent materials and the obtained final materials, thus I think the valence states of these doped elements should also be verified.

Q2. It seems that the undoped sample, NC90, degraded rapidly within the voltage range of 3.0-4.2 V under 25 °C within a pouch cell, much more inferior than that in coin cells, what's the reason accounting for this?

Q3. The authors employed SAED to support the superlattice viewpoint, however, I was not convinced by Figure 4h, which the authors claimed is a result for superlattice. There is no detailed description in the experimental part that how to prepare the TEM sample, if no FIB or ion thinning methods used, the particles might be very thick, thus the diffraction spots might be Secondary diffraction points rather than superlattice. Further, it seems that the diffraction spots for Mo-NC90 is much weaker than Ta-NC90, what's the reason accounting for this?

Reviewer #2 (Remarks to the Author):

This paper discusses the distinguishing factors of dopants optimizing Ni-rich cathode materials through the introduction of Mg, Al, Ti, Ta, and Mo into Li[Ni_{0.91}Co_{0.09}]O₂ cathodes and the investigation of their electrochemical, morphological, and structure properties. The results reveal that the oxidation states of dopants alter the geometry and crystal structure of the cathode grains to influence the cycling performance of cathode materials. Evidently, the dopants with +5 and +6 oxidation states greatly stabilize the micro- and atomic structures of the cathode material as well as the electrochemical performances. This work may have some implications for other researchers in this field. I think this paper could be published in Nature Communication after clarifying the following concerns:

1. Please explain whether the valence state of iron changes after doping. For example, Ti ion has a variety of states, how to ensure that the valence state of Ti doped into the cathode materials is +4. It is recommended to supplement XPS to calibrate the actual valence state after doping.
2. For Li[Ni_{0.91}Co_{0.09}]O₂ cathodes doped with Ta⁵⁺ and Mo⁶⁺, the cycling stability of cathode materials doped with Ta⁵⁺ is relatively better. Additional discussion/evidence should be provided.

3. What is the underlying mechanism mentioned in the text that the doping of high-valent ions, in addition to elongating the shape, can also refine the size of the primary particles?

4. In this paper, the E_{mixing} decreases linearly with increasing oxidation state of the dopant. However, the E_{mixing} of Al^{3+} doping is significantly higher than that of Mg^{2+} doping. Additional discussion should be provided.

Reviewer #3 (Remarks to the Author):

Comments:

This manuscript reports the impact of the oxidation states of dopants (Mg^{2+} , Al^{3+} , Ti^{4+} , Ta^{5+} , and Mo^{6+}) on the electrochemical, morphological, and structural properties of a Ni-rich $Li[Ni_{0.91}Co_{0.09}]O_2$ cathode. However, all of these doping elements have been extensively studied. The author only summarized and compared them in this work under the same doping ratio (1 mol%). The doping ratio of different elements (Mg^{2+} , Al^{3+} , Ti^{4+} , Ta^{5+} , and Mo^{6+}) have different effects on the performance of the material. The low-valent elements (Mg^{2+} , Al^{3+} and Ti^{4+}) doping may also result in excellent performance by increasing the doping ratio. The doping ratio of 1 mol% in this work might just fit for Ta^{5+} , and Mo^{6+} , so the conclusion in this work is not scientific. Thus, it is not innovative enough as a research article to publish in Nature Communications.

Author's Responses to Reviewers' Comments

MS Ref.: NCOMMS-21-23456
Title: **High-energy Ni-rich Layered Cathode Materials for Sustainable Li-ion Batteries**
Authors: H. Hohyun Sun, Un-Hyuck Kim, Jeong-Hyeon Park, Sang-Wook Park, Dong-Hwa Seo, Adam Heller, C. Buddie Mullins, Chong S. Yoon, Yang-Kook Sun

We greatly appreciate the efforts of the reviewers regarding our paper and we have considered and responded to their helpful comments. Indeed, we have reflected their meaningful comments onto the manuscript and conducted additional experiments: (i) XPS verifying the oxidation states of the dopants in cathode materials, (ii) half-cell cycling evaluations of 0.5, 1, and 2 mol % Mg-NC90, Al-NC90, Ti-NC90, Ta-NC90, and Mo-NC90 to determine optimal dopant concentrations, (iii) cross-sectional SEM of 1 and 2 mol % Mg-NC90, Al-NC90, and Ti-NC90 electrodes to observe primary-grain morphologies, and (iv) Rietveld refinement of 0.5, 1, and 2 mol % Mg-NC90, Al-NC90, and Ti-NC90 to observe changes in cation mixing. The reviewers' efforts, suggestions, and criticisms have made our paper much better and we are very thankful for that. Our response is given in blue, and the corrections that we have made are highlighted in yellow in a special version of the revised manuscript (for review only) for the editors and reviewers. We also confirm that all images in figure graphics and TOC graphic are original and have been fully created by the authors.

Reviewers comments:

Reviewer #1 (Remarks to the Author):

The paper submitted by Sun et al. demonstrated that the valence states of the dopants played a key role in enhancing the electrochemical properties of the NC91 materials. The dopants with diverse valence states can alter the morphological properties of the end particles in different levels. The dopants with a high valence state can strengthen the particles, thus leading to improved performance. It's clever that the authors calculated the orientation of grains in each doped particle based on the TEM technique to support their proposal. Besides the morphological viewpoint, a structural view is also claimed. The high valence state Ta⁵⁺ and Mo⁶⁺ can active the formation of superlattice structure, while the dopants with low valence state cannot. This research is complete and I believe it can provide helpful information in guiding the design in the Ni-rich cathode field. However, some points are still confusing.

Authors Response → We appreciate the reviewer for carefully reviewing our manuscript and providing helpful comments. We have revised the manuscript based on these valuable comments and have modified and provided additional information as requested by the reviewer. Furthermore, we have clarified the points that confuse. The corrected parts are marked in yellow in the revised manuscript. We greatly appreciate these comments and the comments provided by the reviewer have improved the quality of the manuscript. We hope that we have provided adequate responses to the comments.

Q1. I was a little confused that why the authors tested the valence states of the elements in dopant parent materials, rather than the final materials. There might be some differences between the valence states in their parent materials and the obtained final materials, thus I think the valence states of these doped elements should also be verified.

Authors Response → Thank you for pointing out the need for this important information. While our original intent was to show that the oxidation state of the dopant parent materials were the intended oxidation states, we agree that it is much more important to verify the oxidation states of the doped elements. Consequently, we have performed XPS on the final materials and analyzed the oxidation states of the doped elements as shown in the modified **Supplementary Fig. 1**. As the modified **Supplementary Fig 1**. shows, the oxidation states of the doped elements are 2+, 3+, 4+, 5+, and 6+, confirming the intended oxidation states of the dopants in the final materials.

The following sentence has been added to the revised manuscript along with the modified **Supplementary Fig 1**: “X-ray photoelectron spectroscopy (XPS) of the dopant parent

materials, MgO, Al₂O₃, TiO₂, Ta₂O₅, and MoO₃, and corresponding doped cathode materials, Mg-NC90, Al-NC90, Ti-NC90, Ta-NC90, and Mo-NC90, confirm that the oxidation states of the dopant metals are +2, +3, +4, +5, and +6, respectively (**Supplementary Fig. 1**).”

These changes are presented on page 5 of the revised manuscript and page 2 of the revised supplementary information.

Supplementary Figure 1. X-ray photoelectron spectroscopy (XPS) results of (a) the dopant parent materials, MgO, Al₂O₃, TiO₂, Ta₂O₅, and MoO₃, and (b) corresponding doped cathode materials, Mg-NC90, Al-NC90, Ti-NC90, Ta-NC90, and Mo-NC90.

Q2. It seems that the undoped sample, NC90, degraded rapidly within the voltage range of 3.0-4.2 V under 25 °C within a pouch cell, much more inferior than that in coin cells, what's the reason accounting for this?

Authors Response → We thank the reviewer for mentioning this point. The reason for the inferior performance of NC90 in pouch-type full-cells, as compared to coin-type half-cells, is the former being subject to higher current densities than the latter. To minimize the capacity loss from the metallic lithium anode, half-cells are constructed with a relatively low cathode active material loading level (4 – 5 mg cm⁻²) and cycled at a low current density (0.5 C-rate, 90 mA g⁻¹) with cycles typically limited to 100 cycles. This is because lithium metal foil, which is used as the anode in half-cells, deteriorates rapidly during cycling from the formation of lithium dendrites and subsequent build-up of dead lithium layer.^{1,2} On the other hand, pouch-type full-cells employ graphite as the anode which is much more stable than Li metal and is constructed with highly cathode active material loading level (10 mg cm⁻²) and cycled at a higher current density (1 C-rate, 200 mA g⁻²), thereby allowing for practical evaluation and testing conditions. Doped cathode materials (Mg-NC90, Al-NC90, Ti-NC90, Ta-NC90, and Mo-NC90) are more resistant to the increase in current density as their crystal structures are reinforced with stabilizing dopants. However, harsher cycling conditions significantly accelerate the degradation rate of the unmodified, fragile NC90 cathode material. Hence, the NC90 cathode material delivers inferior cycling performance in pouch-type full-cells compared to that of coin-type half-cells as seen in **Figures 1b** and **d**.

	Composition of the positive electrode (A.M.: Conducting agent: Binder)	Electrode loading level	Cycling C-rate
Half cell	90: 5.5: 4.5 by wt%	4-5 mg cm ⁻²	0.5 C (90 mA g ⁻¹)
Full cell	94: 3: 3 by wt%	10 mg cm ⁻²	1 C (200 mA g ⁻¹)

[1] J.-Y. Hwang, S.-J. Park, C.S. Yoon, Y.-K. Sun, *Energy Environ. Sci.*, **12**, 2174 (2019)

[2] L. Li, S. Li, Y. Lu, *Chem. Commun.*, **54**, 6648 (2018)

Q3. The authors employed SAED to support the superlattice viewpoint, however, I was not

convinced by Figure 4h, which the authors claimed is an result for superlattice. There is no detailed description in the experimental part that how to prepare the TEM sample, if no FIB or ion thinning methods used, the particles might be very thick, thus the diffraction spots might be Secondary diffraction points rather than superlattice. Further, it seems that the diffraction spots for Mo-NC90 is much weaker than Ta-NC90, what's the reason accounting for this?

Authors Response → Regarding the TEM sample preparation, all of the TEM samples were prepared by FIB and the sample thickness is less than 100 nm which is thin enough for SAED work. As for the extra forbidden spots, if they were generated by secondary (or double diffraction), the spots would immediately disappear if the sample were slightly tilted out of the diffraction zone because the condition for double diffraction is more stringent. The noted diffraction spots in **Figure 5f** (revised version) stayed with the main diffraction spots when tilted, confirming that the extra spots are not from secondary (double) diffraction. Regarding the weak intensity of extra spots for Mo-NC90, the ordered region in that particular primary particle was not as prevalent as other primary particles. The SAED pattern in **Figure 5f** was replaced with a pattern with extra spots with brighter intensity.

The following sentence has been added to the methods section in the revised manuscript regarding TEM sample preparation: “All TEM samples were prepared by using a focusing ion beam (FIB) (NOVA 200/FEI) with sample thicknesses being less than 100 nm for SAED work appropriateness.”

These changes are presented on page 20 of the revised manuscript.

Reviewer #2 (Remarks to the Author):

This paper discusses the distinguishing factors of dopants optimizing Ni-rich cathode materials through the introduction of Mg, Al, Ti, Ta, and Mo into $\text{Li}[\text{Ni}_{0.91}\text{Co}_{0.09}]\text{O}_2$ cathodes and the investigation of their electrochemical, morphological, and structure properties. The results reveal that the oxidation states of dopants alter the geometry and crystal structure of the cathode grains to influence the cycling performance of cathode materials. Evidently, the dopants with +5 and +6 oxidation states greatly stabilize the micro- and atomic structures of the cathode material as well as the electrochemical performances. This work may have some implications for other researchers in this field. I think this paper could be published in Nature Communication after clarifying the following concerns:

Authors Response → We appreciate the reviewer for carefully reviewing our manuscript and providing helpful comments. As the reviewer requested, we revised the manuscript based on these valuable comments and have modified and provided additional information as requested by the reviewer. The modified portions of the manuscript are highlighted in yellow in the revised manuscript. We greatly appreciate these comments and the comments provided by the reviewer have improved the quality of the manuscript. We hope that we have provided adequate responses to the comments.

Q1. Please explain whether the valence state of iron changes after doping. For example, Ti ion has a variety of states, how to ensure that the valence state of Ti doped into the cathode materials is +4. It is recommended to supplement XPS to calibrate the actual valence state after doping.

Authors Response → We appreciate this suggestion and certainly agree that supplementing the manuscript with XPS data to confirm the actual oxidation states of the dopants in the final materials is important. Consequently, we have performed XPS on the final materials and analyzed the oxidation states of the doped elements as shown in the modified **Supplementary Fig. 1**. As the modified **Supplementary Fig. 1** shows, the oxidation states of the doped elements are 2+, 3+, 4+, 5+, and 6+, confirming the intended oxidation states remain unchanged in the final materials.

The following sentence has been added to the revised manuscript along with the modified **Supplementary Fig. 1**: “X-ray photoelectron spectroscopy (XPS) of the dopant parent materials, MgO, Al₂O₃, TiO₂, Ta₂O₅, and MoO₃, and corresponding doped cathode materials, Mg-NC90, Al-NC90, Ti-NC90, Ta-NC90, and Mo-NC90, confirm that the oxidation states of the dopant metals are +2, +3, +4, +5, and +6, respectively (**Supplementary Fig. 1**).”

These changes are presented on page 5 of the revised manuscript and page 2 of the revised

Supplementary Figure 1. X-ray photoelectron spectroscopy (XPS) results of (a) the dopant parent materials, MgO, Al₂O₃, TiO₂, Ta₂O₅, and MoO₃, and (b) corresponding doped cathode materials, Mg-NC90, Al-NC90, Ti-NC90, Ta-NC90, and Mo-NC90.

Q2. For $\text{Li}[\text{Ni}_{0.91}\text{Co}_{0.09}]\text{O}_2$ cathodes doped with Ta^{5+} and Mo^{6+} , the cycling stability of cathode materials doped with Ta^{5+} is relatively better. Additional discussion/evidence should be provided.

Authors Response → We appreciate the reviewer’s comment. In half-cell cycling over 100 cycles at 30 °C, Ta-NC90 shows slightly better performance than Mo-NC90 with capacity retentions of 97.0 and 94.9 %, respectively. However, at extended cycling in pouch-type full-cells, Ta-NC90 and Mo-NC90 deliver 95 and 81.5 % retention rates at the 1000th and 3000th cycles, showing that Ta-NC90 and Mo-NC90 perform almost the same. The difference in retention rates in the half-cells is not significantly higher though, and it could be said that both Ta^{5+} and Mo^{6+} are similarly efficacious in greatly enhancing the long-term electrochemical performance of the NC90 cathode in full cells. Correspondingly, the aspect ratios, relative grain orientations, and grain sizes as well as ordering structure of Ta-NC90 and Mo-NC90 are very similar as seen in the quantitative and crystal structure analysis results of **Fig. 2 and 4**.

Q3. What is the underlying mechanism mentioned in the text that the doping of high-valent ions, in addition to elongating the shape, can also refine the size of the primary particles?

Authors Response → We thank the reviewer for pointing out that further discussion is needed on this topic. The morphologies of the primary particles are determined during the high-temperature calcination process. During this process, the needle-like primary particles of the precursors coalesce into bulkier grains with the insertion of lithium atoms. The presence of Ti, Ta, and Mo dopants helps preserve the original morphology of the precursor primary particles by suppressing the sintering of the primary particles at high temperatures. This results in the characteristically elongated rod-like primary particles seen in the final cathode materials. Furthermore, as shown in **Figure 2g**, the Ta-, Mo-, and Ti-doped cathodes showed much smaller average grain size of primary particle (around 0.3 μm) compared to Al- and Mg-doped cathodes (0.7 μm). Currently, we are conducting further investigations into this aspect and hope provide clear evidences in our next work.

In order to provide further discussion on this topic, we have added the following sentence to the manuscript: “During the 730 °C lithiation process, the needle-like primary particles of the precursors sinter into bulkier grains with the insertion of lithium atoms. The presence of Ti, Ta, and Mo dopants appears to preserve the original morphology of the precursor primary particles by suppressing sintering of the primary particles at 730 °C.”

This change is presented on page 8 of the revised manuscript.

Q4. In this paper, the E_{mixing} decrease linearly with increasing oxidation state of the dopant. However, the E_{mixing} of Al^{3+} doping is significantly higher than that of Mg^{2+} doping. Additional discussion should be provided.

Authors Response → We greatly appreciate the reviewer's valuable comment. The $\text{Li}^+/\text{Ni}^{2+}$ interlayer mixing energy (E_{mixing}) with the dopant substituted instead of Ni^{3+} is mainly determined by two factors: 1) the strength of super-exchange interaction via the $\text{Ni}^{2+}(\text{Li layer, down-spin})-\text{O}^{2-}-\text{Ni}^{x+}$ (TM layer, up-spin) and 2) the lattice strain due to the size difference among the dopant, Li^+ , Ni^{2+} and Ni^{3+} in TM layer. High oxidation state dopants such as Ta^{5+} and Mo^{6+} reduce the neighboring Ni in TM layer from $3+$ to $2+$, resulting in strong super-exchange interaction of $\text{Ni}^{2+}-\text{O}^{2-}-\text{Ni}^{2+}$, decreasing E_{mixing} . In contrast, Mg^{2+} and Al^{3+} do not reduce Ni^{3+} to Ni^{2+} , so $\text{Li}_{32}\text{Ni}_{31}\text{MgO}_{64}$ and $\text{Li}_{32}\text{Ni}_{31}\text{AlO}_{64}$ do not possess strong super-exchange interactions. We have considered all possible dopant sites for Mg^{2+} and selected the most stable configuration among them using DFT calculations; in the most stable $\text{Li}_{32}\text{Ni}_{31}\text{MgO}_{64}$, Mg^{2+} is substituted into the Li layer due to the similar sizes of Mg^{2+} (0.72 Å) and Li^+ (0.76 Å), decreasing its E_{mixing} lower than Al^{3+} 's E_{mixing} . When Mg^{2+} is substituted into the TM layer in $\text{Li}_{32}\text{Ni}_{31}\text{MgO}_{64}$, energy of with Mg^{2+} substituted to the TM layer, the E_{mixing} of Mg^{2+} (0.878 eV) becomes larger than one of Al^{3+} doping (0.778 eV) as the reviewer pointed out. In this case, the difference between Mg^{2+} and Al^{3+} doped structures is the difference in strain between the dopant and Ni^{3+} . The larger E_{mixing} of Mg^{2+} is mainly attributed to the larger strain between Mg^{2+} (0.72 Å) and Ni^{3+} (0.56 Å) than that of Al^{3+} (0.535 Å).

For a systematic comparison, we have revised our manuscript and **Supplementary Figures 20** and **21** along with their caption as shown below.

These changes are presented on pages 21 and 22 of the revised supplementary information.

Supplementary Figure 20. (a) Li/Ni interlayer mixing energy (E_{mixing}) values of $\text{Li}_{32}\text{Ni}_{31}\text{MO}_{64}$ ($M = \text{Mg}^{2+}$, Al^{3+} , Ti^{4+} , Ta^{5+} , and Mo^{6+}) based on a $\text{P2}_1/\text{c}$ LiNiO_2 structure. To compare E_{mixing} systematically, we used the energy of $\text{Li}_{32}\text{Ni}_{31}\text{MgO}_{64}$ with Mg substituted into the TM layer, which is less stable than the model with Mg substituted into the Li layer. (b) $\text{Li}_{32}\text{Ni}_{31}\text{TaO}_{64}$ structure without Li/Ni interlayer mixing (c) $\text{Li}_{32}\text{Ni}_{31}\text{TaO}_{64}$ structure with Li/Ni interlayer mixing. The linear Ni^{2+} (Li layer, down-spin)– O^{2-} – Ni^{2+} (TM layer, up-spin) bonds introduced by Li/Ni interlayer mixing result in strong super-exchange interactions via the σ bonds between the e_g orbitals of Ni^{2+} and the p orbitals of O^{2-} . Note that Mg^{2+} and Al^{3+} do not reduce Ni^{3+} to Ni^{2+} , so $\text{Li}_{32}\text{Ni}_{31}\text{MgO}_{64}$ and $\text{Li}_{32}\text{Ni}_{31}\text{AlO}_{64}$ do not possess strong super-exchange interaction. The difference between Mg^{2+} and Al^{3+} doped structures is the strain difference between the dopant and Ni^{3+} . The larger E_{mixing} of Mg^{2+} is mainly attributed to the larger strain between Mg^{2+} (0.72 Å) and Ni^{3+} (0.56 Å) than that of Al^{3+} (0.535 Å).

Supplementary Figure 21. Li/Ni interlayer mixing energy (E_{mixing}) values of $\text{Li}_{32}\text{Ni}_{31}\text{MO}_{64}$ ($M = \text{Mg}^{2+}, \text{Al}^{3+}, \text{Ti}^{4+}, \text{Ta}^{5+}, \text{and Mo}^{6+}$) based on a C2/m LiNiO_2 structure. For a systematic comparison, we used the energy of $\text{Li}_{32}\text{Ni}_{31}\text{MgO}_{64}$ with Mg substituted into the TM layer, which is less stable than the model with Mg substituted into the Li layer.

Reviewer #3 (Remarks to the Author):

This manuscript reports the impact of the oxidation states of dopants (Mg^{2+} , Al^{3+} , Ti^{4+} , Ta^{5+} , and Mo^{6+}) on the electrochemical, morphological, and structural properties of a Ni-rich $\text{Li}[\text{Ni}_{0.91}\text{Co}_{0.09}]\text{O}_2$ cathode. However, all of these doping elements have been extensively studied. The author only summarized and compared them in this work under the same doping ratio (1 mol%). The doping ratio of different elements (Mg^{2+} , Al^{3+} , Ti^{4+} , Ta^{5+} , and Mo^{6+}) have different effects on the performance of the material. The low-valent elements (Mg^{2+} , Al^{3+} and Ti^{4+}) doping may also result excellent performance by increasing the doping ratio. The doping ratio of 1 mol% in this work might just fit for Ta^{5+} , and Mo^{6+} , so the conclusion in this work is not scientific. Thus, it is not innovative enough as a research article to publish in *Nature Communications*.

Authors Response → We thank the reviewer for carefully reviewing our manuscript and regret that the quality of the manuscript was not up to par with the reviewer's expectations. We recognize the validity of the reviewer's comments in stating the need to test the doping ratios of the different materials since increased doping ratios may result in excellent performances for the lower oxidation state dopant-based materials. Based on this valuable suggestion, we have performed further experiments to strengthen our manuscript. Specifically, we have carried out half-cell cycling evaluations of differing doping concentrations (0.5, 1, and 2 mol %) of Mg^{2+} , Al^{3+} , Ti^{4+} , Ta^{5+} , and Mo^{6+} , cross-sectional SEM observations of 2 mol % Mg-NC90, Al-NC90, Ti-NC90 electrodes, and Rietveld refinement of 2 mol % Mg-NC90, Al-NC90, Ti-NC90 XRD patterns.

The half-cell cycling evaluations verify that best cycling performance are obtained by the Mg-NC90, Al-NC90, Ti-NC90, Ta-NC90, and Mo-NC90 materials when the dopant concentration is 1 mol %, confirming that the optimal dopant concentration is 1 mol % (**Supplementary Figure 3**). Furthermore, cross-sectional SEM images of 1 and 2 mol % Mg-NC90, Al-NC90, and Ti-NC90 and Rietveld refinement of 0.5, 1, and 2 mol % Mg-NC90, Al-NC90, and Ti-NC90 XRD patterns confirm that increases in dopant concentrations do not alter the primary-grain morphologies as well as the cation mixing value (**Figures R1 and R2** shown below). We hope that the additional experiments that we have conducted to address the reviewer's concerns will satisfy the reviewer's view on the scientific quality of the manuscript.

The following sentence has been added to the revised manuscript along with **Supplementary Fig. 3**: "Note that the half-cell cycling performances of differing dopant concentrations (0.5, 1, and 2 mol %) of Mg^{2+} , Al^{3+} , Ti^{4+} , Ta^{5+} , and Mo^{6+} in **Supplementary Fig. 3** show that optimal cycling performances are shown at 1 mol % dopant concentration."

These changes are presented on page 6 of the revised manuscript and page 4 of the revised supplementary information.

Supplementary Figure 3. Half-cell cycling of differing dopant concentrations (0.5, 1, and 2 mol %) of Mg²⁺, Al³⁺, Ti⁴⁺, Ta⁵⁺, and Mo⁶⁺: (a, c, e, g, i) first charge-discharge cycle curves

at $0.1\text{ C} = 18\text{ mA g}^{-1}$ and (b, d, f, h, j) cycling at $0.5\text{ C} = 90\text{ mA g}^{-1}$ over 100 cycles at $30\text{ }^{\circ}\text{C}$ and between $2.7 - 4.3\text{ V}$ voltage window.

Figure R1. Cross-sectional SEM images of as-synthesized cathode particles show the primary-grain morphologies of Mg-NC90, Al-NC90, and Ti-NC90 with 1 and 2 mol % dopant concentrations. The SEM images show that an increase in dopant concentration to 2 mol % does not alter the grain morphologies.

Figure R2. (a) Degree of cation mixing through the interchange of Li^+ and Ni^{2+} of Mg-NC90, Al-NC90, and Ti-NC90 with 0.5, 1, and 2 mol % dopant concentrations which demonstrate that cation mixing amount does not change significantly. (b) Corresponding Rietveld refined XRD patterns of 0.5, 1, and 2 mol % Mg-NC90, Al-NC90, and Ti-NC90.

REVIEWER COMMENTS

Reviewer #1 (Remarks to the Author):

Thanks for the revise made by the authors. However, I was not persuaded by the XPS results added by the author (shown in Figure S1). There are significant differences in the binding energy of the doped elements before and after sintering, which I consider cannot be assigned to the same valence states. I recommend other techniques be performed regarding this matter.

Reviewer #2 (Remarks to the Author):

This paper discusses the distinguishing factors of dopants optimizing Ni-rich cathode materials through the introduction of Mg, Al, Ti, Ta, and Mo into $\text{Li}[\text{Ni}_{0.91}\text{Co}_{0.09}]\text{O}_2$ cathodes and the investigation of their electrochemical, morphological, and structure properties. The results reveal that the oxidation states of dopants alter the geometry and crystal structure of the cathode grains to influence the cycling performance of cathode materials. Evidently, the dopants with +5 and +6 oxidation states greatly stabilize the micro- and atomic structures of the cathode material as well as the electrochemical performances. This work may have some implications for other researchers in this field. I think this paper could be published in Nature Communication.

Author's Responses to Reviewers' Comments

MS Ref.: NCOMMS-21-23456

Title: **High-energy Ni-rich Layered Cathode Materials for Sustainable Li-ion Batteries**

Authors: H. Hohyun Sun, Un-Hyuck Kim, Jeong-Hyeon Park, Sang-Wook Park, Dong-Hwa Seo, Adam Heller, C. Buddie Mullins, Chong S. Yoon, Yang-Kook Sun

We greatly appreciate the efforts of the reviewers regarding our paper and we have considered and responded to their helpful comments. Indeed, the reviewers' efforts, suggestions, and criticisms have improved the quality of our paper and we are thankful for that. We have reflected the reviewers' meaningful comments onto the manuscript and reconducted the XPS experiment of the doped cathode materials with sputtered surfaces to exclude the influence from the complex chemical environment present on cathode material surface. Our response is given in blue, and the corrections that we have made are highlighted in yellow in a special version of the revised manuscript (for review only) for the editors and reviewers. We also confirm that all images in figure graphics and TOC graphic are original and have been fully created by the authors.

Reviewers comments:

Reviewer #1 (Remarks to the Author):

Thanks for the revise made by the authors. However, I was not persuaded by the XPS results added by the author (shown in Figure S1). There are significant differences in the binding energy of the doped elements before and after sintering, which I consider cannot be assigned to the same valences. I recommend other techniques be performed regarding this matter.

Authors Response → We thank the reviewer for the valuable comment and for pointing out the disparities between the XPS spectra (previous **Supplementary Fig. 1**) of the dopant parent materials and the corresponding doped materials. This is an important point as in XPS, the differences in binding energy (BE) peak locations can suggest different levels of oxidation states. Still, one also has to keep in mind that the BE peak positions are heavily influenced by the local chemical environment, and this is especially true for LIB materials whose surfaces are chemically complex.¹⁻⁴ XPS BE peaks are often calibrated using the C1s peak from the adventitious carbon-based contaminant, which has a BE of 284.8 eV, as a reference. However, due to the electronically insulating nature of multiple of chemical phases relevant to battery interfaces, such as Li-associated phases, phase identification reliance primarily on absolute XPS core-level BEs is difficult as sample surface charging effects from insulating materials shift core-levels by several eVs during XPS measurements.⁵ For Ni-rich layered oxide cathodes, such insulating phases are mainly represented by residual lithium (e.g. LiOH and Li₂CO₃) which transforms the surface chemical bonding and electrical properties of NCM material surface. As a result, transition metal dopant BE peak shapes and locations can change under such circumstances. For example, Verdier *et al.* compared the XPS spectra of Al₂O₃ and LiAl_{0.15}Co_{0.85}O₂ and reported that the Al 2p peaks of Al³⁺ in a pristine and in a battery surface environment differs.⁶ **Revision Figure 1** shows that the sharp Al 2p peak of Al₂O₃ at broadens and shifts to a lower BE. The differences in XPS spectra of Al-doped pristine layered oxide cathodes are also supported by Aurbach *et al.* and Huang *et al.* where the Al 2p peak was reported to appear as a broad peak in the 73.0 – 74.5 eV BE region.^{7,8}

The shift of the XPS spectrum occurs because photoelectrons ejected from an electronically insulating sample leave behind a charge, leading to shift of the XPS spectrum. Taking note of this, we retook XPS measurements of the dopant parent materials (MgO, Al₂O₃, TiO₂, Ta₂O₅, and MoO₃,) and the corresponding doped cathode materials (Mg-NC90, Al-NC90, Ti-NC90, Ta-NC90, and Mo-NC90) but used an electron flood gun to neutralize the surface charge. Additionally, we sputtered the surface of the cathode samples with a 500 eV Ar⁺ ion etching gun to remove ~ 8 nm of cathode surface to minimize the influence of the local chemical environment on the XPS spectra. As a result, The peak location and shapes of the XPS spectra of the dopant parent materials (new **Supplementary Fig. 1a**) and doped cathode materials (new **Supplementary Fig. 1b**) match, evidencing that the oxidation states of Mg, Al, Ti, Ta, and Mo

are 2+, 3+, 4+, 5+, and 6+, respectively.

The previous **Supplementary Figure 1** has been replaced with the new **Supplementary Figure 1** (shown below). We have also updated the Methods section to provide a more detailed XPS experimental with the following: “The oxidation states of dopants in cathodes and relative Ni²⁺ and Ni³⁺ contents (Ni²⁺ or Ni³⁺/(Ni²⁺+Ni³⁺), expressed as a percentage) of the cathode surfaces were determined via XPS (Axis Supra, Kratos/PHI 5000, Versaprobe). Data was taken using a monochromatic Al-K α X-ray source (15 kV, 300W) with a spot size of 400 x 400 μm^2 . Dopant oxidation states in cathodes were measured with the electron flood gun on and the cathode surface was etched with a Ar⁺ 500 eV ion etching gun prior to taking measurements.”

These changes are presented on page 19 of the revised manuscript and page 2 of the revised supplementary information.

Revision Figure 1. XPS Al 2p-Co3p-Li 1s core peaks for pristine LiCoO₂, Al₂O₃-coated

LiCoO₂, AlPO₄-coated LiCoO₂, AlPO₄-coated LiCoO₂, LiAl_{0.15}Co_{0.85}O₂, and Al₂O₃. Adapted from reference 10.

Supplementary Figure 1. X-ray photoelectron spectroscopy (XPS) results of (a) the dopant parent materials, MgO, Al₂O₃, TiO₂, Ta₂O₅, and MoO₃, (b) corresponding doped cathode

materials, Mg-NC90, Al-NC90, Ti-NC90, Ta-NC90, and Mo-NC90, and (c) survey spectra of the doped cathode materials.

References

1. Tonti, D.; Pettenkofer, C.; Jaegermann, W. Origin of the Electrochemical Potential in Intercalation Electrodes: Experimental Estimation of the Electronic and Ionic Contributions for Na Intercalated into TiS₂. *J. Phys. Chem. B* **2004**, *108*, 16093-16099.
2. Lindgren, F.; Rehnlund, D.; Källquist, I.; Nyholm, L.; Edstrom, K.; Hahlin, M.; Maibach, J. Breaking Down a Complex System: Interpreting PES Peak Positions for Cycled Li-Ion Battery Electrodes. *J. Phys. Chem. C* **2017**, *121*, 27303-27312.
3. Oswald, S. Binding Energy Referencing for XPS in Alkali Metal- Based Battery Materials Research (I): Basic Model Investigations. *Appl. Surf. Sci.* **2015**, *351*, 492-503.
4. Oswald, S.; Hoffmann, M.; Zier, M. Peak Position Differences Observed during XPS Sputter Depth Profiling of the SEI on Lithiated and Delithiated Carbon-Based Anode Material for Li-Ion Batteries. *Appl. Surf. Sci.* **2017**, *401*, 408-413.
5. Wood, K. N.; Teeter, G. XPS on Li-Battery-Related Compounds: Analysis of Inorganic SEI Phases and a Methodology for Charge Correction. *ACS Appl. Energy Mater.* **2018**, *1*, 4493-4504.
6. Verdier, S.; El Ouatani, L.; Dedryvere, R.; Bonhomme, F.; Biensan, P.; Gonbeau, D. XPS Study on Al₂O₃- and AlPO₄-Coated LiCoO₂ Cathode Material for High-Capacity Li Ion Batteries. *J. Electrochem. Soc.* **2007**, *154*, A1088-A1099.
7. Aurbach, D.; Srur-Lavi, O.; Ghanty, C.; Dixit, M.; Haik, O.; Talianker, M.; Grinblat, Y.; Leifer, N.; Lavi, R.; Major, D. T.; Goobes, G.; Zinigrad, E.; Erickson, E. M.; Kosa, M.; Markovsky, B.; Lampert, J.; Volkov, A.; Shin, J.-Y.; Garsuch, A. Studies of Aluminum-Doped LiNi_{0.5}Co_{0.2}Mn_{0.3}O₂: Electrochemical Behavior, Aging, Structural Transformations, and Thermal Characteristics. *J. Electrochem. Soc.* **2015**, *162*, A1014-A1027.
8. Huang, J.; Du, K.; Peng, Z.; Cao, Y.; Xue, Z. Duan, J.; Wang, F.; Liu, Y.; Hu, G. Enhanced High-Temperature Electrochemical Performance of Layered Nickel-Rich Cathodes for Lithium-Ion Batteries after LiF Surface Modification. *ChemElectroChem* **2019**, *6*, 5428-5432.

REVIEWERS' COMMENTS

Reviewer #1 (Remarks to the Author):

The paper can be accepted for publication. Thanks for your clarification in the XPS results.

Author's Responses to Reviewers' Comments

MS Ref.: NCOMMS-21-23456B

Title: **Transition metal-doped Ni-rich layered cathode materials for durable Li-ion batteries**

Authors: H. Hohyun Sun, Un-Hyuck Kim, Jeong-Hyeon Park, Sang-Wook Park, Dong-Hwa Seo, Adam Heller, C. Buddie Mullins, Chong S. Yoon, Yang-Kook Sun

We appreciate the efforts of the reviewer regarding our paper and giving us their final comment.

Reviewers comments:

Reviewer #1 (Remarks to the Author):

The paper can be accepted for publication. Thanks for your clarification in the XPS results.

Authors Response → Thank you!